# Pyridine Derivatives—A New Class of Compounds That Are Toxic to *E. coli* K12, R2–R4 Strains

**DOI:** 10.3390/ma14185401

**Published:** 2021-09-18

**Authors:** Dominik Koszelewski, Ryszard Ostaszewski, Paweł Śmigielski, Anastasiia Hrunyk, Karol Kramkowski, Łukasz Laskowski, Magdalena Laskowska, Rafał Lizut, Mateusz Szymczak, Jacek Michalski, Kamil Gawin, Paweł Kowalczyk

**Affiliations:** 1Institute of Organic Chemistry PAS, Kasprzaka 44/52, 01-224 Warsaw, Poland; d.koszelewski@icho.edu.pl (D.K.); r.ostaszewski@icho.edu.pl (R.O.); p.smigielski@icho.edu.pl (P.Ś.); a.hrunyk@icho.edu.pl (A.H.); 2Department of Physical Chemistry, Medical University of Bialystok, Kilińskiego 1 Str., 15-089 Białystok, Poland; kkramk@wp.pl; 3Institute of Nuclear Physics Polish Academy of Sciences, 31-342 Krakow, Poland; lukasz.laskowski@ifj.edu.pl (Ł.L.); magdalena.laskowska@ifj.edu.pl (M.L.); 4The John Paul II Catholic University of Lublin, Institute of Mathematics, Informatics and Landscape Architecture ul. Konstantynów 1 H, 20-708 Lublin, Poland; lizut@kul.pl; 5Department of Molecular Virology, Institute of Microbiology, Faculty of Biology, University of Warsaw, Miecznikowa 1, 02-096 Warsaw, Poland; mszymczak@biol.uw.edu.pl; 6Department of Animal Nutrition, The Kielanowski Institute of Animal Physiology and Nutrition, Polish Academy of Sciences, 05-110 Jabłonna, Poland; j.michalski@ifzz.pl (J.M.); k.gawin@ifzz.pl (K.G.)

**Keywords:** pyridine derivatives, *E. coli* strains, oxidative stress, antibiotics, lipopolysaccharide

## Abstract

A preliminary study of 2-amino-4-aryl-3,5-dicarbonitrile-6-thiopyridines as new potential antimicrobial drugs was performed. Special emphasis was placed on the selection of the structure of target pyridine derivatives with the highest biological activity against different types of Gram-stained bacteria by lipopolysaccharide (LPS). Herein, *Escherichia coli* model strains K12 (without LPS in its structure) and R2–R4 (with different lengths of LPS in its structure) were used. Studied target compounds were provided with yields ranging from 53% to 91% by the lipase-catalyzed one pot multicomponent reaction of various aromatic aldehydes with malononitrile, and thiols. The presented work showed that the antibacterial activity of the studied pyridines depends on their structure and affects the LPS of bacteria. Moreover, the influence of the pyridines on bacteria possessing smooth and rough LPS and oxidative damage to plasmid DNA caused by investigated compounds was indicated. Additionally, the modification of the bacterial DNA with the tested compounds was performed to detect new potential oxidative damages, which are recognized by the Fpg protein. The obtained damage modification values of the analyzed compounds were compared with the modifications after antibiotics were used in this type of research. The presented studies demonstrate that 2-amino-4-aryl-3,5-dicarbonitrile-6-thiopyridines can be used as substitutes for known antibiotics. The observed results are especially important in the case of the increasing resistance of bacteria to various drugs and antibiotics.

## 1. Introduction

The pyridine ring system can be found in a number of natural products and in several pharmacologically relevant compounds. Among others, 2-amino-4-aryl-3,5-dicarbonitrile-6-thiopyridines have gained considerable attention due to their wide-ranging biological activities (Figure 1) [1].

These pyridine derivatives were found to inhibit PrPSc accumulation in scrapie-infected mouse neuroblastoma cells (ScN2a) [2], act as an adenosine A1 receptor agonist for the chronic treatment of heart diseases [3,4] modulate androgen receptor function [5], and function as IKK-2 for treating HBV infection [6], Additionally, these compounds are often used as anti-prion [7,8], anti-hepatitis B virus [9], anti-bacterial [10], and anti-cancer [11] agents, and as potassium channel openers for the treatment of urinary incontinence [12]. Recently, these compounds have been recognized as potential targets for the development of new drugs for the treatment of Alzheimer and neuronal vascular diseases, Parkinson’s, hypoxia, asthma, kidney, epilepsy, and Creutzfeldt–Jacob diseases [13,14,15,16]. At present, there are insufficient data in the professional scientific literature on the cytotoxic properties of simple synthetic pyridine derivatives performed on *Escherichia coli* R1–R4 model bacterial cells. Investigating pyridine derivatives and their potentially toxic effect on bacterial cells may reveal them be helpful in many new cases of drug resistance (including antibiotics) as effective antimicrobials against bacterial clinical pathogens. The aim of the study is to analyze the potential antibacterial activity of 2-amino-4-aryl-3,5-dicarbonitrile-6-sulfanylpyridines as potential antibiotics in terms of their structure and their cytotoxic effect on bacterial lipopolysaccharides (LPS) in the model strains used in this type of research on *Escherichia coli* K12 (without LPS in its structure) and R2–R4 (LPS of different lengths in its structure).

## 2. Materials and Methods

### 2.1. General Methods of Synthesis Pyridine Derivatives

All the chemicals were purchased from Sigma-Aldrich and the solvents were of analytical grade. Melting points of all the synthesized compounds were determined in open capillary tubes and are uncorrected. Toluene was distilled over Na/K-alloy in the presence of benzophenone. ^1^H NMR spectra were recorded in CDCl_3_ or DMSO-*d*_6_ solution using Varian Gemini 400 NMR spectrometer (400 MHz). MS spectra were reordered on API 3000 mass spectrometer (Applied Biosystems). TLC analyses were performed on Kieselgel 60 F_254_ aluminum sheets. TLC were developed in KMnO_4_ solution containing K_2_CO_3_ and NaOH. Lipases from porcine pancreas, Type II (PPL) (catalogue number L-3126, Lot. Number 108H1379), *Pseudomonas fluorescens* (PFL) (catalogue number 534730, Lot. Number MKBH1198V), *Candida rugosa* (CRL) (catalogue number 90860, Lot. Number BCBH7102V), *Candida cylindracea* (CCL) (catalogue number 62316, Lot. Number 1336707), *Burkholderia ÿepacian* (BCL) (catalogue number 534641, Lot. Number MKBV0029V) and *Rhizomucor miehei* (RML) (catalogue number 80484) and Bovine serum albumin were purchased from Sigma-Aldrich. Immobilized lipase from *Candida antarctica* B (Novozym 435) (catalogue number LC200223) was purchased from Novo Nordisk. Enzymatic reactions were performed in a vortex (Heidolph Promax 1020) equipped with incubator (Heidolph Inkubator 1000). To prove the ability of the established protocol, each reaction was repeated at least three times.

### 2.2. General Procedure for the Synthesis of 2-Amino-4-aryl-3,5-dicyano-6-phenyl Thiopyridines (***5a***–***k***)

A mixture of an aldehyde (1 mmol), PPL (100 mg), malononitrile (2 mmol) and thiol (1 mmol) in ethanol (2 mL) was shaken at 200 rpm at 40 °C for 18 h. Reaction was terminated by filtering off the catalyst through the bed of celite. The MCR toward **5a**, catalyzed by different enzymes are recorded in Table 1. The melting points and the yields of the derivatives are recorded in Table 2. Melting points and spectral data of **5a**–**k** remained in agreement with the literature data. The structure of all compounds was confirmed using NMR and mass spectroscopy and, for some compound, the correct elemental analysis data were also recorded in each method synthesis.


**2-Amino-4-(4-cyanophenyl)-6-(4-methylphenylsulfanyl)-3,5-pyridinedicarbonitrile (5a)**




Compound **5a** obtained according to general method with 83% yield (305 mg, 0.83 mmol) as white crystal, mp 275 °C (MeOH) [Lit. mp 272–274 °C, EtOH]; ^[1] 1^H NMR (400 MHz, DMSO-*d*_6_) δ 8.06 (d, *J* = 8.4 Hz, 2H), 7.85 (s, 2H), 7.77 (d, *J* = 8.4 Hz, 2H), 7.46 (d, *J* = 8.2 Hz, 2H), 7.29 (d, *J* = 7.9 Hz, 2H), 2.35 (s, 3H). ^1^H NMR data were in accordance with those reported in the literature. ^[2]^ LRMS m/z (ESI) calc. for C_21_H_13_N_5_SNa [M + Na]^+^: 390.0. Found: 390.0.


**2-Amino-4-(thiophen-2-yl)-6-(4-methylphenylsulfanyl)-3,5-pyridinedicarbonitrile (5b)**




Compound **5b** obtained according to general method with 78% yield (271 mg, 0.78 mmol) as yellow crystal, mp 197 °C (MeOH) [Lit. mp 196–198 °C, MeOH]; ^[2] 1^H NMR (400 MHz, DMSO-*d*_6_) δ 8.00–7.89 (m, 1H), 7.77 (s, 2H), 7.55 (dd, *J* = 3.7, 1.3 Hz, 1H), 7.45 (d, *J* = 8.1 Hz, 2H), 7.36–7.23 (m, 3H), 2.35 (s, 3H); ^1^H NMR data were in accordance with those reported in the literature. ^[2]^ LRMS m/z (ESI) calc. for C_18_H_13_N_4_S_2_ [M + H]^+^: 349.0, Found: 349.0.


**2-Amino-6-((4-aminophenyl)thio)-4-phenylpyridine-3,5-dicarbonitrile (5c)**




Compound **5c** obtained according to general method with 52% yield (178 mg, 0.52 mmol) as yellow crystal, mp 223 °C (EtOH) [Lit. mp 219–221 °C]; ^[3] 1^H NMR (400 MHz, DMSO-*d*_6_) δ 7.67 (s, 2H), 7.58–7.47 (m, 5H), 7.17 (d, *J* = 8.6 Hz, 2H), 6.61 (d, *J* = 8.6 Hz, 2H), 5.56 (s, 2H). ^1^H NMR data were in accordance with those reported in the literature. ^[3]^ LRMS m/z (ESI) calc. for C_19_H_14_N_5_S [M + H]^+^: 344.1, Found: 344.1.


**2-Amino-4-phenyl-6-(*p*-tolylthio)pyridine-3,5-dicarbonitrile (5d)**




Compound **5d** obtained according to general method with 89% yield (304 mg, 0.89 mmol) as white crystal, mp 249 °C (EtOH) [Lit. mp 248–250 °C (EtOH)]; ^[4] 1^H NMR (400 MHz, CDCl_3_) δ 7.53 (hept, *J* = 3.5, 3.0 Hz, 5H), 7.43 (d, *J* = 8.2 Hz, 2H), 7.27–7.21 (m, 2H), 5.46 (s, 2H), 2.42 (s, 3H).^1^H NMR data were in accordance with those reported in the literature. ^[5]^ LRMS m/z (ESI) calc. for C_20_H_15_N_4_S [M + H]^+^: 343.0, Found: 343.0.


**2-Amino-6-(octylthio)-4-phenylpyridine-3,5-dicarbonitrile (5e)**




Compound **5e** obtained according to general method with 76% yield (277 mg, 0.76 mmol) as white crystal, mp 148 °C (EtOH) [Lit. mp 151–153 °C (EtOH)]; ^[6] 1^H NMR (400 MHz, CDCl_3_) δ 7.73–7.37 (m, 5H), 5.63 (s, 2H), 3.20 (t, *J* = 7.3 Hz, 2H), 1.73 (t, *J* = 7.5 Hz, 2H), 1.45 (t, *J* = 7.5 Hz, 2H), 1.40–1.21 (m, 8H), 0.89 (t, *J* = 7.5 Hz, 3H). ^1^H NMR data were in accordance with those reported in the literature. ^[6]^ LRMS m/z (ESI) calc. for C_21_H_25_N_4_S [M + H]^+^: 365.1, Found: 365.1.


**2-Amino-4-(p-tolyl)-6-(p-tolylthio)pyridine-3,5-dicarbonitrile (5f)**




Compound **5f** obtained according to general method with 91% yield (324 mg, 0.91 mmol) as white crystal, mp 223–224 °C (EtOH) [Lit. mp 222–224 °C (EtOH)]; ^[1] 1^H NMR (400 MHz, DMSO-*d*_6_) δ 7.68 (s, 2H), 7.47 (dd, *J* = 14.3, 8.4 Hz, 4H), 7.33–7.24 (m, 2H), 7.10 (d, *J* = 8.8 Hz, 2H), 3.83 (s, 3H), 2.35 (s, 3H). ^1^H NMR data were in accordance with those reported in the literature. ^[1]^ LRMS m/z (ESI) calc. for C_21_H_17_N_4_S [M + H]^+^: 357.1, Found: 357.1.


**2-Amino-6-((4-chlorophenyl)thio)-4-phenylpyridine-3,5-dicarbonitrile (5g)**




Compound **5g** obtained according to general method with 89% yield (322 mg, 0.89 mmol) as white crystal, mp 247 °C (EtOH) [Lit. mp 245–247 °C]; ^[6] 1^H NMR (400 MHz, DMSO-*d*_6_) δ 7.82 (s, 2H), 7.70–7.33 (m, 9H). ^1^H NMR data were in accordance with those reported in the literature. ^[6]^ LRMS m/z (ESI) calc. for C_19_H_12_ClN_4_S [M + H]^+^: 363.0, Found: 363.0.


**2-Amino-4-(4-methoxyphenyl)-6-(p-tolylthio)pyridine-3,5-dicarbonitrile (5h)**




Compound **5h** obtained according to general method with 82% yield (305 mg, 0.82 mmol) as white crystal, mp 230–231 °C (EtOH) [Lit. mp 229–231 °C]; ^[4] 1^H NMR (400 MHz, CDCl_3_) δ 7.50 (d, *J* = 8.8 Hz, 2H), 7.42 (d, *J* = 8.1 Hz, 2H), 7.25 (d, *J* = 8.1 Hz, 2H), 7.04 (d, *J* = 8.8 Hz, 2H), 5.46 (s, 2H), 3.87 (s, 3H), 2.41 (s, 3H). ^1^H NMR data were in accordance with those reported in the literature. ^[4]^ LRMS m/z (ESI) calc. for C_21_H_17_N_4_OS [M + H]^+^: 373.1, Found: 373.1.


**2-Amino-4-(4-nitrophenyl)-6-(p-tolylthio)pyridine-3,5-dicarbonitrile (5i)**




Compound **5i** obtained according to general method with 88% yield (341 mg, 0.88 mmol) as white crystal, mp 298 °C (EtOH) [Lit. mp 300–302 °C]; ^[7]^ 1H NMR (400 MHz, DMSO-*d*_6_) δ 8.25 (d, *J* = 8.7 Hz, 2H), 7.70 (d, *J* = 8.4 Hz, 2H), 7.28 (d, *J* = 8.4 Hz, 2H), 7.15 (d, *J* = 7.7 Hz, 2H), 2.25 (s, 3H). ^1^H NMR data were in accordance with those reported in the literature. ^[7]^ LRMS m/z (ESI) calc. for C_20_H_14_N_5_O_2_S [M + H]^+^: 388.1, Found: 388.1.


**2-Amino-6-((4-bromophenyl)thio)-4-phenylpyridine-3,5-dicarbonitrile (5j)**




Compound **5j** obtained according to general method with 74% yield (301 mg, 0.74 mmol) as white crystal, mp 255–257 °C (EtOH) [Lit. mp 256–258 °C]; ^[3] 1^H NMR (400 MHz, DMSO-*d*_6_) δ 7.83 (s, 2H), 7.67 (d, *J* = 8.1 Hz, 2H), 7.55 (dd, *J* = 11.3, 5.6 Hz, 7H). ^1^H NMR data were in accordance with those reported in the literature. ^[3]^ LRMS m/z (ESI) calc. for C_19_H_12_BrN_4_S [M + H]^+^: 406.9, Found: 406.9.


**2,4-Diamino-5-[(4-methylphenyl)thio]-5H-[1]benzopyrano[2,3-b]pyridine-3-carbonitrile (5k)**




Compound **5k** obtained according to general method with 79% yield (285 mg, 0.79 mmol) as white crystal, mp 224 °C (EtOH) [Lit. mp 223–225 °C]; ^[8] 1^H NMR (400 MHz, DMSO-*d*_6_) δ 7.17 (d, *J* = 7.4 Hz, 2H), 7.07 (td, *J* = 7.2, 6.6, 1.2 Hz, 1H), 6.96–6.84 (m, 4H), 6.80 (d, *J* = 1.0 Hz, 1H), 6.63 (d, *J* = 8.1 Hz, 2H), 6.43 (s, 2H), 5.67 (s, 1H), 2.20 (s, 3H). ^1^H NMR data were in accordance with those reported in the literature. ^[8]^ LRMS m/z (ESI) calc. for C_20_H_17_N_4_OS [M + H]^+^: 361.1, Found: 361.1.

### 2.3. Microorganisms and Media

All bacterial strains were received and statistically analysed, as described in [5,6,7,8,9,10,11,12,13,14,15,16,17,18,19,20,21,22,23,24,25,26,27,28,29,30,31,32,33,34,35,36,37,38,39,40,41,42,43,44,45,46,47,48,49,50,51,52,53,54,55].

## 3. Results

### 3.1. Chemistry

Heterocyclic compounds are the largest class of organic compounds. Many natural products and most drugs have heterocyclic rings. Antibiotics, dyes of flowers and other plants, compounds that transport oxygen to various organs of our body, and DNA components are heterocyclic compounds. They can be of different sizes, have multiple bonds, and can be linked by chains or rings. Among the heterocyclic compounds, heterocyclic nitrogen and oxygen compounds deserve special attention. The synthesis of pyridine derivatives, aiming to develop new drugs, is an active research area. The general method for the preparation of 2-amino-4-aryl-3,5-dicarbonitrile-6-sulfanylpyridines is based on a multicomponent condensation reaction (MCR) between aldehydes, malononitrile (two equiv.), and thiols in the presence of a catalyst [17,18]. Various catalysts have been reported to effect MCR, including nano-crystalline magnesium oxide, [19] silica nanoparticles [20], ZnCl_2_ [21], piperidine/microwave [22] and Cd(II) metal–organic frameworks [23], refluxed in basic alumina [24]. Although the reported methods are efficient to provide the desired 2-amino-4-aryl-3,5-dicarbonitrile-6-sulfanylpyridines, there are still some drawbacks regarding these protocols, such as their longer reaction time, limited substrate scope, and complicated work-up processes, as well as the application of expensive, highly toxic and carcinogenic transition metal catalysts. Therefore, the development of a catalyst system that does not contain harmful components such as transition metals, strong acids or bases seems desirable.

As a part of our ongoing research into environmentally sustainable protocols [25,26,27,28], we devoted our attention to the development of a protocol for the desired 2-amino-4-aryl-3,5-dicarbonitrile-6-sulfanylpyridines, which combines both economic and green chemistry aspects. Although several methods have recently been reported regarding the synthesis of 2-amino-4-aryl-3,5-dicarbonitrile-6-sulfanylpyridines [37,38,39,40,41,42,43,44], the biocatalytic approach to target compounds is strictly limited to the application of a whole cell of Baker yeast [29]. As a result of our intensive work on the development of an effective method for the synthesis of pharmaceutically relevant compounds [30,31], we wish to report hydrolases as a sustainable catalyst for MCR, leading to the target pyridines **5** [32,33,34,35].

Based on our previous studies on promiscuous enzyme activity [34] the model MCR of 4-cyanobenzaldehyde (1 mmol), malononitrile (2 mmol) and 4-methylthiophenol was used in cyclohexane at 40 ◦C (Figure 2), (Table 1, entry 1).

To screen the suitable medium, we performed the model reaction in different solvents: water, ethanol, acetonitrile, *N*,*N*-dimethyl formamide (DMF) and dimethyl sulfoxide (DMSO). The obtained results are summarized in Table 1. Protic solvents such as H_2_O and EtOH were found to enhance the reaction yield of the product **5a**. The obtained results remained in agreement with the literature data [29]. To evaluate the effect of temperature, the model reaction was carried out at a temperature that was increased to 50 °C; however the yield was reduced above 40 °C, providing product **5a** with 75%, which may be explained by the changes in the tertiary structure of the used enzyme (Table 1, entry 2, 8, 9).

We applied the developed reaction conditions using various aldehydes and thiophenols, which resulted in substituted pyridines **5b**–**k** with high yields (Table 2). The structures of the obtained products **5** were verified by a comparison of spectral data (NMR and mass) and melting point temperatures with the literature data for title compounds. The characterization data of the synthetized compounds **5a**–**k** are presented in the experimental part.

### 3.2. Toxicity of Piryidine Derivatives

In our current studies into the toxicity of pyridine derivatives on model *E. coli* bacterial cells, we used MIC and MBC tests, similar to our earlier experiments [45,46,47,48,49,50]. They consisted of examining the rate of migration of the analyzed compounds through the bacterial membranes that may be damaged by them, which, in turn, may lead to apoptosis of the bacterial cell. This is especially visible when the color of the analyzed sample changes after the addition of a special resazurin dye, which changes its color from dark blue to pink, orange or yellow [45,46,47,48,49,50,51,52]. As a result of irreversible damage to the bacterial membrane by a given compound, the dye penetrates with varying intensity, but only reaches dead cells (Appendix A).

The experiments were carried out using new methods of synthesizing 11 new compounds–pyridine derivatives **5** (Figure 1 and Figure 2, Table 1 and Table 2), containing basic three, four or more fused aromatic rings in their basic structure, connected by various functional groups, such as sulfur (S), nitrogen (N), amino (NH_2_), cyanide (CN), chlorine (Cl), bromine (Br), methyl (OMe) and nitrogen (NO_2_), at the R_1_ or R_2_ position. The research investigated the influence of these groups, which are attached to the rings of pyridine derivatives (compounds **5a**, **5b**, **5c**, **5d**, **5e**, **5f**, **5g**, **5h**, **5i**, **5j**, **5k**).

The toxic effect on bacterial cells after the analysis of the MIC and MBC test for all 11 analyzed compounds was obtained for the compounds marked in our study, with symbols as in **5a** and **5g**–**5k,** for which the MIC values were observed in the range of 0.2–1.3 µg/mL, and 10–42 µg/mL for MBC values in the analyzed model strains K12, R2, R3 and R4) (Figure 3 and Figure 4), which had specific functional groups in the structure of the R_1_ and R_2_ substituent. The antimicrobial activity of the selected pyridine derivatives was determined on the basis of the MBC values for the analyzed compounds, which ranged from 30 to 45 µg/mL (Figure 5). The indicator showed an increase in the value from about 160 to 250 µg/mL in *E. coli* R2–R4 strains compared to the K12 strain, where, in all the analyzed MIC and MBC tests, due to the lack of LPS in its structure (Figure 3, Figure 4 and Figure 5), values remained at the very low level of slightly above zero (0.12–0.21 µg/mL for MIC and 1–4 µg/mL for MBC), (Figure 3 and Figure 4) and (0–25 µg/mL for MBC/MIC) (Figure 5).

In the analyzed MIC plates, after using all the analyzed compounds (Appendix A), for the reference *E. coli* strain K12, a color change was observed at a dilution of 10-6, which corresponds to an MIC value of 0.003225 μg/mL^−1^ in all analyzed reactions. On panel B, a dilution in an R2 strain was on the level of 10-3, especially for compounds marked as 1,3,7,8,9,10,11 (corresponding to **5a**, **5c**, **5g**–**5k**), containing substituents such as CN, S and Br. The MIC values for these compounds were 0.027 µg/mL^−1^. For the remaining analyzed compounds in the R_2_ substituent, the color change was visible at a dilution of 10-5, which corresponds to an MIC value of 0.00525 µg/mL^−1^. The observed color change was more intense than in the other analyzed compounds. This proves the permanent and irreversible destruction of the bacterial membrane, and the LPS of a certain length contained in it, by the analyzed compounds in the R2 skeleton. This was observed in the R3 and R4 strains. The relationships between the analyzed strains and compounds are shown in Figure 3, Figure 4 and Figure 5 and in Appendix A and in Table 3.

### 3.3. Modification of Plasmid DNA Isolated from E. coli R2–R4 Strains with Tested Pirydine Derivatives

Based on the results of MIC and MBC toxicity tests in the analyzed model of bacterial (Appendix A), it was decided to modify the bacterial DNA with the analyzed pyridine compounds. On the basis of the obtained results for the modification of bacterial DNA, we observed that, in the R4 strain, there was a very distinct change in the structure caused by changing the ratio of the topological forms of the ccc, linear and oc plasmid DNA to each other, and the formation of densely looped structures forming the so-called concatamers, the formation of which may be caused by bacterial DNA topoisomerases (Appendix A). No such visible and significant changes in the structure were observed in the bacterial DNA obtained and modified from other strains, K12, R2 and R3. For further analysis with the Fpg protein (which is a bifunctional glycosylase and recognizes oxidized DNA bases such as 8oxoG, FapyA and FapyG) [45,46,47,48,49,50], we used plasmid DNA. Since we observed the greatest number of modifications after treatment with the Fpg protein in the R4 strain, we present its values as an exemplary model (Appendix A). As the greatest damage, after modification with pyridine derivatives and antibiotics after digestion with Fpg protein, was visible in the R4 strain, we present it as an example in the Appendix A.

After treatment with Fpg glycosase in strain R4, we observed clearly visible damage in the topological changes in plasmid DNA forms; “ccc”, linear form and “oc”, which have been completely damaged and are only visible as stray bands (see Figure 6, Appendix A) according to the earlier literature (data [45,46,47,48,49,50]).

The next step in our research was to use pyridine derivatives as a function of the commonly used beta-lactam antibiotics in the treatment of specific bacterial infections, which include kanamycin, streptomycin, ciprofloxacin, bleomycin and cloxacillin, a component of the drug syntarpen. They are mainly used in skin and soft tissue infections, lower respiratory tract infections and osteomyelitis. As in our previous experiments [45,46,47,48], we also used three antibiotics as a reference compounds in the present study to test their effect on the analysed pyridine derivatives (Figure 7). The experimental system used was identical to the analyzed compounds [45,46,47,48] with the use of MIC and MBC tests in terms of quantity and concentration (Appendix A) and for bacterial DNA isolated from the bacteria modified by these antibiotics and digested with protein Fpg. We observed that, in all analyzed R-type strains, based on the MIC and MBC tests, there was a visible color change in the analyzed antibiotics used at a dilution of 10-3 (Appendix A), which corresponds to the values of 0.027 µg/mL^−1^ in the analyzed MIC (Appendix A). It was also shown that the R4 strain, having the longest LPS, interacts with all active groups included in the analyzed antibiotics. The obtained results were also statistically significant at the level of *p* < 0.05. In the analyzed pyridine derivatives, especially in compounds **1**,**3**,**7**–**10** (which corresponds to the determinations in graphs **5a**, **5c**, **5g**–**5k**), the MIC values were similar to the values in the model strain R4, which proves that these compounds can also potentially be used as “alternatives” to commonly used antibiotics.

In all analyzed nucleic acids after modification with antibiotics and digestion with Fpg protein, changes in all topological forms were **observed** in various proportions (Appendix A). These lesions were similar to those observed after modification with selected pyridine derivatives and digestion with Fpg protein, but significantly weaker (Figure 8). This may indicate that the modification with an appropriate antibiotic in bacterial DNA constituted additional new substrates recognized by the Fpg protein, whose structure was similar to that of the pyridine derivatives also recognized by this protein.

In the nucleic acids isolated from the R4 strain after modification with cloxacillin and digestion with the Fpg protein, two additional forms appeared over the traditional topological plasmids ccc and oc, which indicates the formation of additional looped structures in the bacterial genetic material. This effect was especially visible after modification with 1,3,7–10 compounds containing C, S, CN, Br as a substituent function. In the remaining analyzed samples, digested with the same antibiotic and not digested with the Fpg protein, the occurrence of all topological forms of the plasmid was unchanged. After digestion of these samples, trailing bands of topological plasmid forms appeared, which were rearranged as a result of digestion with the Fpg protein, showing oxidative base modifications in bacterial DNA as “smear”. Their intensity varied depending on the pyridine derivative used (Appendix A).

The highest values (ranged from 2.7 to 3.3%) of the observed damage in plasmid DNA were observed for the pyridine derivatives numbered 1,3,7–10, and similarly for the samples modified with cloxacillin, in which the R4 strain was the highest (R4 > R2 > R3 > K12), as in our earlier studies [45,46,47,48].

This proves that they damage the DNA of the bacterial cell more than the analyzed antibiotics by interfering with its structure (disturbed structural topological forms of bacterial DNA). This indicates a very high toxicity of the analyzed pyridine derivatives towards bacterial DNA, caused by the significant modification of the components of the bacterial membrane and the LPS contained in it, which may activate bacterial topoisomerases, allowing for the relaxation of the structure and access to the modified, exposed DNA bases.

Presumably, stabilization of the topoisomerase-controlling complex is necessary for the cell to survive. Blocking these enzymes blocks replication and transcription, which can affect the total amount of super-replicated DNA.

## 4. Discussion

The toxicity of selected pyridine derivatives on model strains of *E. coli* containing LPS -R2–R4 in their structure (or a lack of K12 strain) determines them as potentially new drug precursors to commonly and traditionally used antibiotics [37,38,39,40,41,42,43,44].

The use of model strains of *E. coli* makes it possible to understand the exact mechanism of the destruction of the bacterial cell membrane and the changes in the redox potential of its individual components.

All 11 compounds used on model bacterial cells were highly toxic, especially the seven containing CN, NO_2_ groups and S, N, Br atoms in certain positions in the R_1_ and R_2_ substituents. These showed the highest antibacterial effectiveness. The highest values in both types of MIC and MBC assay, and the highest percentage of modifications identified after digestion with Fpg protein, were observed for compounds marked as 1, 3, 7–11 (corresponding to the determinations in compounds **5a**, **5c**, **5g**–**5k**, respectively). This indicates that the type of the substituent and specific number of aromatic rings determine its toxic effect on model *E. coli* bacterial cells that differ in LPS length in ascending order R2 < R3 < R4. The analysis of the literature data shows that the analyzed strains can cause civilization diseases related to the dysfunction of various organs and tissues in the human body [45,46,47,48,49,50,51,52].

The effect of the analysed compounds on the bacterial membrane was very similar to the interaction of ionic liquids containing quaternary ammonium surfactants and other analyzed compounds, such as coumarin derivatives, 1,2-diaaryl alcohols and lactones [45,46,47,48,49,50,51,52].

Changes in the activity of bacterial cells under the influence of pyridine derivatives probably result from the strong activation of oxidative stress in the bacterial cell at its cell membrane, which, at a later stage, leads to apoptosis as a result of further penetration into the genetic material. This leads to its destruction through the modification and inhibition of the replication apparatus leading to apoptosis [50,51,52,53,54,55].

The obtained results constitute the basis for the continuation of research into other bacterial strains associated with human civilization diseases and identification of hospital infections. In addition to the MIC and MBC tests, the analyzed pyridine derivatives were analyzed for the digestion of modified plasmids isolated from strains K12 and R2, R3 and R4 by the Fpg protein with N-glycosylase/AP lyase activity [50,51,52,53,54,55].

The number of damaged plasmids which were modified by pyridine derivatives was determined on the basis of the rearrangement of topological changes in bacterial DNA forms additionally digested with Fpg glycosylase. On the basis of numerous literature reports, it was found that the Fpg protein has a very wide-ranging removal of modified bases as a result of oxidation and alkylation processes, with the active participation of free oxygen radicals induced by ROS or RNS. At present, it is believed in many domestic and foreign laboratories that this protein is an extremely precise and very sensitive marker of DNA oxidative damage formed in the bacterial cell in the process of oxidative stress induced by factors of internal origin, e.g., lipid peroxidation, alkylation, methylation of DNA bases, etc., as well as external origin such as ionizing radiation. It is estimated that the amount of oxidation in bacterial DNA at the level of 3–4%, caused by the aforementioned factors and recognized by the Fpg enzyme, is an important indicator of the modification of individual DNA bases, including guanine, leading to the formation of 8oxoguanine (8oxoG), FapyGua (FapyG) or modification of adenine to form Fapy Ade (FapyA) in a particular matrix [45,46,47,48,49,50,51,52,53,54,55].

In studies with pyridine derivatives, changes in the topological forms of bacterial DNA were observed for seven of the 11 analyzed compounds after digestion with Fpg protein. These changes consisted of mixing all forms in the form of “smearing” bands after digestion with Fpg protein, and modified bacterial plasmids. This suggests that these are new potential substrates for this protein [45,46,47,48,49,50,51,52,53,54,55].

The obtained results show that the analyzed pyridine derivatives significantly modify the bacterial DNA isolated from the model strains of *E. coli*, K12 and R. These modifications are recognized by the Fpg protein (Appendix A). The most effective pyridine derivatives were compounds **5a**, **5c**, **5g**–**5k**. This means that, in the future, some of these analyzed compounds could be modified to be innovative, less toxic to the human body and more toxic to bacterial cells. They could be considered as potential drugs, and used as substitutes for antibiotics with a similar chemical structure. It is now known that the constant and frequent use of antibiotics results in the acquisition of immunity for many bacteria, including pathogenic ones [53]. Therefore, in our experiments we tested five commonly used antibiotics—kanamycin, streptomycin, cloxacycline, blemycin and ciprofloxacin—which are used in patients with cardiovascular and digestive system infections caused by staphylococci of the genus *Stahylococcus areus* [45,46,47,48,49,50]. These antibiotics have a very broad spectrum of activity against Gram-negative bacteria, including *E. coli* [45,46,47,48,49,50], as demonstrated on the respective MIC and MBC plates (Appendix A). As in our earlier studies, after modifying the bacterial DNA with a selected antibiotic and digesting it with the Fpg protein, the highest level of damage was observed for the strains R4 > R2 > R3 > K12. These values were twice as high for the respective pyridine derivatives as compared to the antibiotics used on bacterial DNA in all analyzed strains, especially in the R4 strain (Appendix A). Therefore, in our methods of innovative syntheses, we are looking for compounds with a similar structure to antibiotics, but with much stronger biological and chemical properties, which will be more toxic to the analyzed bacterial cells [45,46,47,48,49,50,51,52,53,54,55], (Table 2).

In bacterial DNA isolated from the R4 strain, digested with modified pyridine derivatives, and appropriate antibiotics digested with the Fpg protein, a disappearance of the “ccc” form was observed, along with the appearance of a single-strand migration, similar to the “oc” form, which formed a complex with high molecular weight (see Appendix A). The effect of the separation and migration of bands in both experiments was very similar, and confirms that selected pyridine derivatives will replace commonly used antibiotics in the future, enhancing their biological functions by affecting the bacterial cell membrane. This is suggested by the results of digestion with the Fpg protein (Appendix A), which forms a strong complex in the form of visible concatamers after modification of bacterial DNA. The resulting complex is very stable in terms of electrophoresis. This suggests a covalent linkage of the Fpg protein and the DNA through appropriate amino acids or high-molecular-weight protein fragments, as well as the resulting DNA damage after modification with pyridine derivatives and antibiotics [45,46,47,48,49,50,51,52,53,54,55].

In previous work, a similar mechanism of action was described for the Fpg protein [45,46,47,48,49,50]. The analyzed model *E. coli* strains, including the particularly sensitive R4 strain, showed greater sensitivity than the R2 and R3 strains, as they had the longest LPS (Appendix A). Toxicity studies of the synthesis of new compounds and their effect on bacterial cells will allow for the better selection of substituents, leading to the creation of modern drugs with ideal biological parameters and microbial activity for the analyzed bacterial cells.

## 5. Conclusions

A new enzymatic method of target pyridine derivatives, which eliminates the application of toxic catalysts, was developed. For the first time, the promiscuous activity of lipases was shown in a multicomponent reaction, leading to the highly functionalized bioactive 2-amino-4-aryl-3,5-dicarbonitrile-6-sulfanylpyridines in organic solvent. The presented protocol offers several advantages, such as operational simplicity, environmental sustainability, easy work-up procedure, and high yields of the target products (53–91%). The toxic effect of the obtained pyridine derivatives was evaluated on model *E. coli* strains. The analyzed pyridine derivatives are able to modify all *E. coli* model strains (R2–R4) and their bacterial DNA, changing the spatial structure of the LPS contained in their cell membranes. Among the derivatives studied, the compounds **5b**, **5e**, **5f** and **5j** were the most active. The activity of the tested compounds **5** strongly depends on the structure of the R_2_ substituent of the thiol unit (Figure 2). It was also noted that the change in the bromine atom in the aromatic ring to chlorine significantly reduced the activity of the compound **5g**. Therefore, the studies will be continued in our laboratory to determine the influence of the R_2_ unit structure of pyridine derivatives on their antibacterial activity. The toxicity of the aromatic groups in the analyzed pyridine derivatives, along with the alkyl substituents in the R_1_ and R_2_ positions, probably depends on their interaction with the membrane, which may be involved in the destruction of the cell walls by changing their hydrophobicity. Changes in the permeability and integrity of the bacterial membrane may result in a specific bacterial response to biologically active compounds, e.g., the antibiotics used. The damage of plasmid DNA bases, which is especially visible after digestion with Fpg protein, was associated with alkylation and oxidative modifications induced by pyridine derivatives. This may suggest that the presence of these compounds has a toxic effect on bacterial LPS, generating strong oxidative stress, as was observed in our previous studies [45,46,47,48,49,50,51,52]. The results of the presented research are important for understanding the biological properties of tested pyridine derivatives as a function of potential new antibiotics and their toxic effects on Gram-negative bacteria in the face of the growing drug-resistance pandemic. This can be seen in our previous works, related to the characteristics of the model *E. coli* K12 and R2–R4 [44,45,46,47,48,49,50,51]. Finally, the analyzed pyridine derivatives are more cytotoxic in the model bacterial cells than the most commonly used antibiotics: kanamycin, streptomycin, ciprofloxacin, bleomycin and cloxacillin.

## Figures and Tables

**Figure 1 materials-14-05401-f001:**
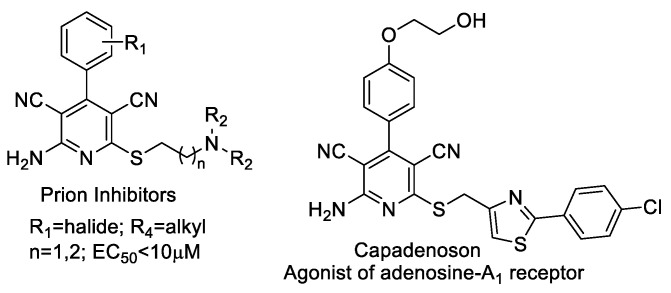
The 2-amino-4-aryl-3,5-dicarbonitrile-6-sulfanylpyridines as medicinally privileged structures.

**Figure 2 materials-14-05401-f002:**
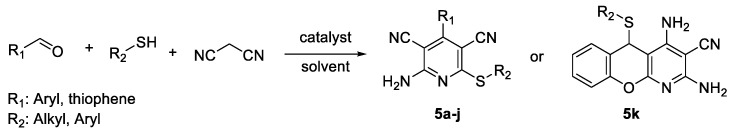
Enzyme-catalyzed synthesis of pyridines **5a**–**k**.

**Figure 3 materials-14-05401-f003:**
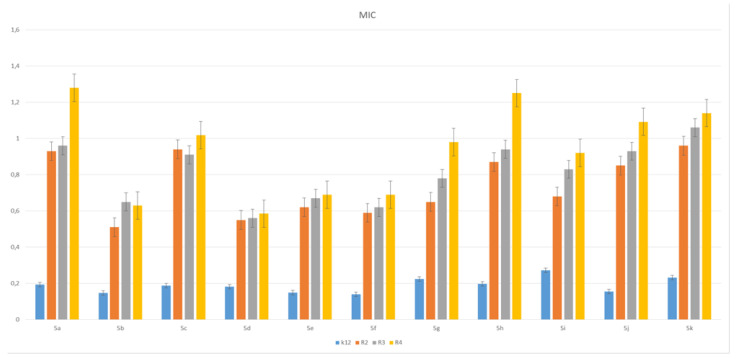
Minimum inhibitory concentration (MIC) of the pyridine derivatives in model bacterial strains. The *x*-axis 11 compounds were used sequentially. The *y*-axis shows the MIC value in µg/mL^−1^.

**Figure 4 materials-14-05401-f004:**
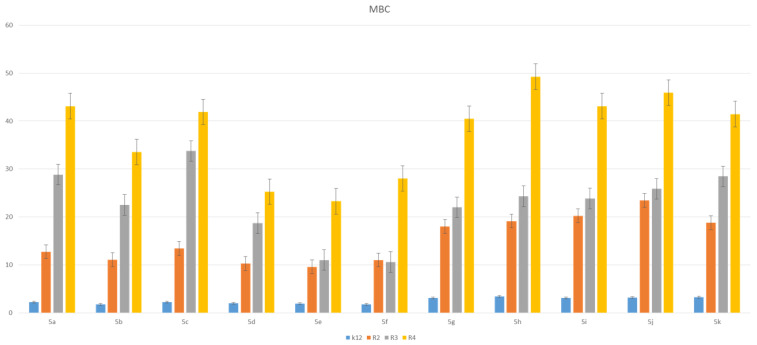
MBC of the pyridine derivatives in model bacterial strains. The *x*-axis 11-compounds were used sequentially. The *y*-axis shows the MBC value in µg/mL^−1^.

**Figure 5 materials-14-05401-f005:**
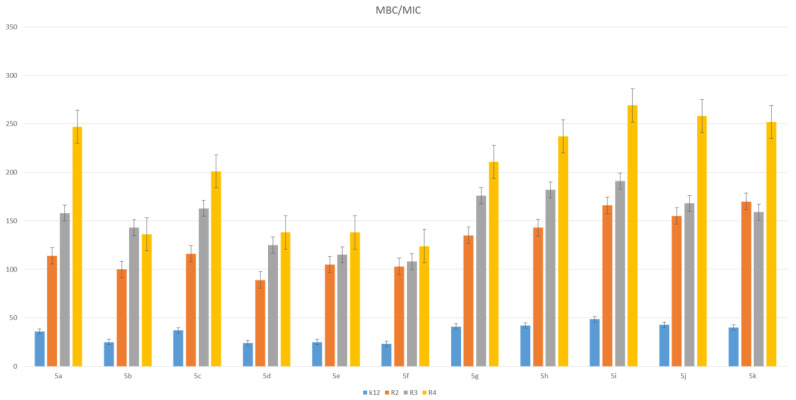
MBC/MIC of the pyridine derivatives in model bacterial strains. The *x*-axis compounds **1**–**11** were used sequentially. The *y*-axis shows the MBC/MIC value in µg/mL^−1^.

**Figure 6 materials-14-05401-f006:**
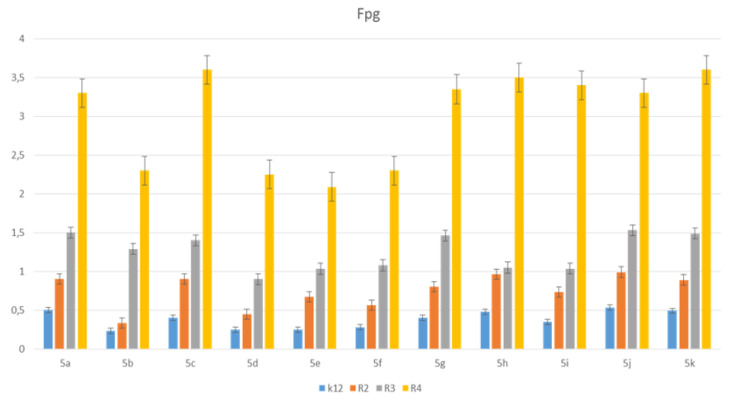
Percent of digested DNA damages recognised by Fpg enzyme- (*y*-axis) with control K12 and R2–R4 strains (*x*-axis); The selected compounds **1**–**17** were statistically significant at *p* < 0.05 *.

**Figure 7 materials-14-05401-f007:**
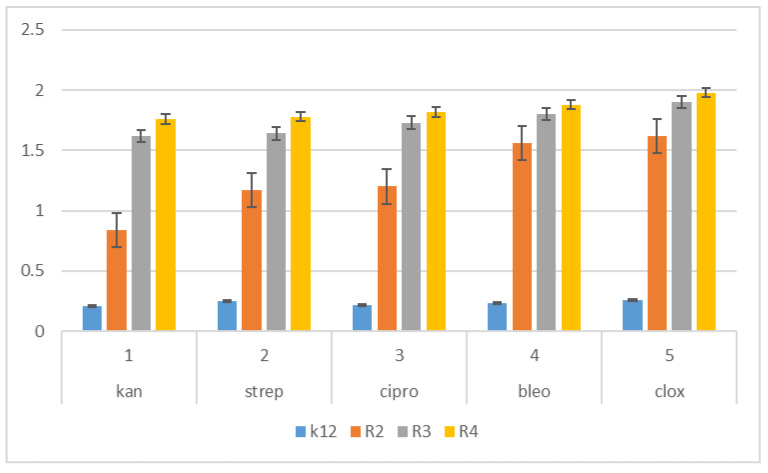
Examples of MIC with model bacterial strains K12, R2, R3, and R4 of the studied antibiotics with kanamycin, streptomycin, ciprofloxacin, bleomycin and cloxacillin. The *x*-axis features antibiotics that were used sequentially. The *y*-axis features the MIC value in µg/mL^−1^.

**Figure 8 materials-14-05401-f008:**
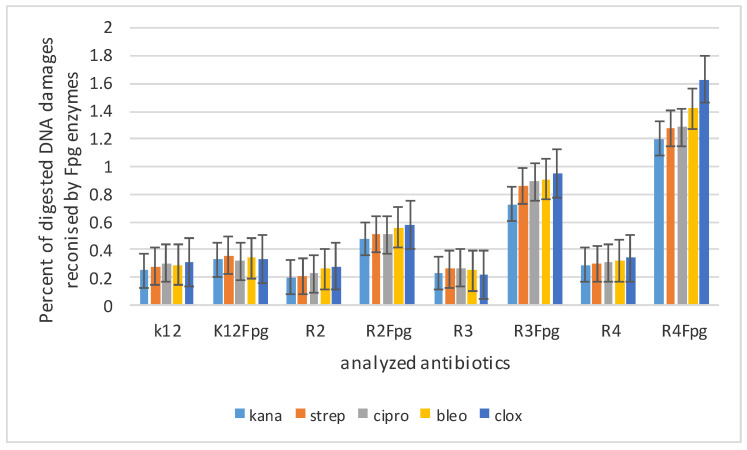
Percentage of bacterial DNA recognized by Fpg enzyme in model bacterial strains after kanamycine, sterpromycine, ciprofloxacin, bleomycine and cloxsacilline treatment. The compounds were statistically significant at *p* < 0.05.

**Table 1 materials-14-05401-t001:** The MCR toward **5a**, catalyzed by different enzymes ^[a]^.

Entry	Catalyst	T (°C)	Solvent	Yield [%] ^e^
1	None	40	cyclohexane	4
2	Porcine pancreas lipase (PPL)	40	cyclohexane	31
3	Porcine pancreas lipase (PPL)	40	H_2_O	49
4	Porcine pancreas lipase (PPL)	40	EtOH	83
5	Porcine pancreas lipase (PPL)	40	MeCN	52
6	Porcine pancreas lipase (PPL)	40	DMF	24
7	Porcine pancreas lipase (PPL)	40	DMSO	33
8	Porcine pancreas lipase (PPL)	50	EtOH	75
9	Porcine pancreas lipase (PPL) ^b^	40	EtOH	80
10	*Pseudomonas fluorescens* lipase (PFL)	40	cyclohexane	11
11	*Candida rugosa* lipase (CRL)	40	cyclohexane	16
12	*Candida cylindracea* lipase (CCL)	40	cyclohexane	18
13	Novozym 435	40	cyclohexane	4
14	Bovine serum albumin (BSA)	40	cyclohexane	8
15	Pig liver acetone powder (PLAP) ^c^	40	cyclohexane	14
16	Denatured PPL ^d^	40	cyclohexane	6

^a^ Reaction conditions: 4-cyanobenzaldehyde (1 mmol), malononitrile (2 mmol) and 4-methylthiophenol, and enzyme (100 mg) in solvent (2 mL) for 18 h, 200 rpm. ^b^ PPL (120 mg), ^c^ Domestically prepared. ^d^ Thermally deactivated. ^e^ Yield of the isolated product 5a, [29,36].

**Table 2 materials-14-05401-t002:** Yields of reaction provided for isolated products 5 ^a^.

Entry	Compound	R_1_	R_2_	Yield [%] ^b^	Mp (°C) ^c^
1	5a	4-CNC_6_H_4_	4-MeC_6_H_4_	83	275
2	5b	thiophene	4-MeC_6_H_4_	78	197
3	5c	Ph	4-NH_2_C_6_H_4_	53	223
4	5d	Ph	4-MeC_6_H_4_	89	249
5	5e	Ph	*n*-C_6_H_13_	76	148
6	5f	4-MeC_6_H_4_	4-MeC_6_H_4_	91	223–224
7	5g	Ph	4-ClC_6_H_4_	89	247
8	5h	4-MeOC_6_H_4_	4-MeC_6_H_4_	82	230–231
9	5i	4-NO_2_C_6_H_4_	4-MeC_6_H_4_	88	298
10	5j	Ph	4-BrC_6_H_4_	74	255–257
11	5k	2-OHC_6_H_4_	4-MeC_6_H_4_	79	224

^a^ Reaction conditions: Aldehyde (1 mmol), malanonitrile (2 mmol), thiol (1 mmol), PPL (100 mg), ethanol (2 mL), stirred at 40 °C for 18 h. ^b^ Yield of isolated product. ^c^ Melting points remain in agreement with those reported in the literature (see Appendix A).

**Table 3 materials-14-05401-t003:** Statistical analysis of all 11 analyzed compounds at *p* < 0.05 *, <0.01 **, <0.001 *** in MIC, MBC and MBC/MIC tests. Pyridine derivatives 1–11 were used sequentially.

No of Samples	1	2	3	4	5	6	7	8	9	10	11	Type of Test
K12	***	*	**	*	*	*	**	***	*	**	**	MIC
R2	***	*	**	*	*	*	**	***	*	***	**	MIC
R3	***	*	**	*	*	*	**	***	*	**	**	MIC
R4	***	*	**	*	*	*	**	***	*	**	**	MIC
K12	***	*	**	*	*	*	**	**	*	**	***	MBC
R2	**	*	**	*	*	*	**	**	*	**	***	MBC
R3	**	*	**	*	*	*	**	**	*	**	***	MBC
R4	**	*	**	*	*	*	**	**	*	**	***	MBC
K12	*	*	**	*	*	*	*	*	*	**	*	MBC/MIC
R2	*	*	*	*	*	*	*	*	*	**	*	MBC/MIC
R3	*	*	*	*	*	*	*	*	*	**	*	MBC/MIC
R4	*	*	*	*	*	*	*	*	*	**	*	MBC/MIC

## Data Availability

On request of those interested.

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
