# Peer review of "Pyridine Derivatives—A New Class of Compounds That Are Toxic to E. coli K12, R2–R4 Strains"

_materials, 2021, doi:10.3390/ma14185401_

Round 1
Reviewer 1 Report
The work presents an analysis of the effect of different synthetized chemical compounds on different bacteria strains. The topic is interesting. however, there is no material perspective in the manuscript. Furthermore, Its quality is below the standards of a scientific work. Some specific comments
The term pyridine scaffold appears as not suitable in the manuscript. A scaffold is a supramolecular structure and in the manuscript It appears as a simple group.
Introduction should be rewritten in a more concise way. In the current form, the connection between the different discussed aspects remains unclear
Clear information of the sources of the used chemicals is required
Section 2.1 does not introduce any general method for pyridine synthesis
Section 3.1 does not present any result, It only includes some general aspects related to the Chemistry of the compounds
It is not clear wether authors studied pyridine or pyrimidine derivates
The meaning of K12, R2,R3 and R4 should be clearly stated in the manuscript.
Discussion does not state clearly the differences between the different strains or compounds
Conclusions appear as a summary of findings but not real conclusions are included in the manuscript.
Author Response
Reviewer1
Comments and Suggestions for Authors
The work presents an analysis of the effect of different synthetized chemical compounds on different bacteria strains. The topic is interesting. however, there is no material perspective in the manuscript. Furthermore, Its quality is below the standards of a scientific work. Some specific comments
Dear Reviewer,
Thank you very much for the thorough review and valuable comments. Please find bellow our response to your comments.
Reviewer 1
The term pyridine scaffold appears as not suitable in the manuscript. A scaffold is a supramolecular structure and in the manuscript It appears as a simple group.
Response
We are grateful to the Reviewer for this suggestion. It was modified.
Reviewer 1
Introduction should be rewritten in a more concise way. In the current form, the connection between the different discussed aspects remains unclear
Response
We are grateful to the Reviewer for this suggestion. The Introduction was modified.
Reviewer 1
Clear information of the sources of the used chemicals is required
Response
Due to the Reviewer suggestion the information of the sources of the used chemicals was provided.
Reviewer 1
Section 2.1 does not introduce any general method for pyridine synthesis
Response
Yes we agree with the Reviewer, the general method for pyridine derivatives synthesis is provided in Section 2.2 entitled General procedure for the synthesis of 2-amino-4-aryl-3,5-dicyano-6-phenyl thiopyridines (5a–k).
Reviewer 1
Section 3.1 does not present any result, It only includes some general aspects related to the Chemistry of the compounds
Response
We are grateful to the Reviewer for this suggestion. According to the Reviewer suggestion Section 3.1 was modified and supplemented with results from chemical studies under synthesis of target pyridines.
Reviewer 1
It is not clear wether authors studied pyridine or pyrimidine derivates
Response
We fully agree with the Reviewer. The research was devoted to pyridine derivatives. As suggested by the reviewer, this has been corrected along the manuscript.
Reviewer 1
The meaning of K12, R2,R3 and R4 should be clearly stated in the manuscript.
Discussion does not state clearly the differences between the different strains or compounds
Conclusions appear as a summary of findings but not real conclusions are included in the manuscript.
Response
The differences between the used model E.coli strains that do not have LPS, such as the K12 strain, or have LPS of increasing R2-R4 length, have been thoroughly described in our previous works [44-49 and 50-51]. Here they have been described signaling in order to avoid potential self plagiarism in sentence sequences or phrases very similar to each other describing the given words together with their exact structure and LPS characteristics. Therefore the chapter conclusions summarizes it exactly, based on our earlier descriptions of the LPS in the works, which was also included in the manuscript

Reviewer 2 Report
The manuscript describes the development of a new class of pyridine derivatives that show antibacterial activity against model strains. The synthesis of those compounds is catalysed by porcine pancrease lipase and is a green process with high reaction yields. The study reports a substantial body of work that can be of interest for the readers of the journal and the wider community. The manuscript is well organised and is easy to follow. I suggest the following modifications:
1.The protocol followed for the determination of MIC and MBC tests should be moved to section 2.
2. The conclusion section should be more compact.
3. Are there any limitations/drawbacks associated with the proposed synthetic protocol?
4. The authors should expand on the green/environmental aspect of their study.
Author Response
Dear Reviewer,
Thank you very much for the thorough review and valuable comments. Please find bellow our response to your comments.
Reviewer 2
1.The protocol followed for the determination of MIC and MBC tests should be moved to section 2.
Response
Has been included in section 2.3. materials and methods
Reviewer 2
- The conclusion section should be more compact.
Response
The conclusions have been modified
Reviewer 2
- Are there any limitations/drawbacks associated with the proposed synthetic protocol?
Response
The high substate selectivity observed for the enzymes used as a catalyst may influence the reaction yields. Due to the high substrate selectivity, it is sometimes necessary to find the appropriate enzyme for each substrate individually to reach product with high yield. New catalytic activity of lipases has been presented toward synthesis of target pyridine derivatives. To our knowledge, this phenomenon has not been observed for this class of enzymes in the case of the synthesis of target compounds. Therefore, a number of additional studies are required to provide a more precise answer to the potential limitations of this method.
Reviewer 2
- The authors should expand on the green/environmental aspect of their study.
Response
We are grateful for this suggestion. Application of enzymes as catalysts eliminates the need of using toxic and cancerogenic catalysts.

Reviewer 3 Report
In this article, Kowalczyk and co-workers described an enzymatic catalytic MCR strategy to synthesize series of 2-amino-4-aryl-3,5-dicarbonitrile-6-sulfanylpyridine derivatives. In addition, the compounds they've obtained were subjected to series of biological experiments for their antibiotic activities. Overall, the work is sound, and the grammar is fine with some minor mistakes. However, the major issue that I have for this article is their work may not fit the aim or topics of Materials. Therefore, I would suggest the authors consider subject their work to other journals (such as Molecules or BMC).
It is suggested to the authors to submit their work to appropriate journals for the right audience.
Author Response
Dear Reviewer,
Thank you very much for the thorough review and valuable comments. Please find bellow our response to your comments.
Reviewer 3
Overall, the work is sound, and the grammar is fine with some minor mistakes. However, the major issue that I have for this article is their work may not fit the aim or topics of Materials. Therefore, I would suggest the authors consider subject their work to other journals (such as Molecules or BMC).
It is suggested to the authors to submit their work to appropriate journals for the right audience.
Response
We are grateful for this suggestion. However the studies described in the manuscript is in line with the research topic published in Materials (Materials 2020, 13, 2499; doi:10.3390/ma13112499; Materials 2020, 13, 5169; doi:10.3390/ma13225169; Materials 2021, 14, 1025. https://doi.org/10.3390/ma14041025; Materials 2021, 14, 2956. https://doi.org/10.3390/ma14112956

Reviewer 4 Report
In this manuscript Submitted to Special Issue "Analysis of Peptidomimetic Toxicity in E. coli Bacterial Cells—Studies on Selected Strains" authors present an enzymatic catalytic protocol to synthesize Pyridine derivatives and evaluate their toxicity against E. coli K12, R2-R4 Strains. It is no doubt an interesting issue, and in the scope of the journal, however in my opinion, authors should make some reorganization before publication.
Comment 1) Introduction should not contain results. Data presented in Table 1 and 2 should be relocated to the section 3. Results in a subtitle 3.x. Catalytic Synthesis (for instance, or similar).
Comment 2) Scheme 1, mentioned in line 80, do not appear in the text.
Comment 3) Rewrite the sentence of lines 81 -83.
Comment 4) Please verify the use of the word pyridine and pyrimidine all along the text. I believe that sometimes are improperly used.
Comment 5) Why authors used 100 % of organic solvent, as reaction medium? Under these conditions, it may be expected low yields due to enzyme inactivation.
Comment 6) Why present melting point values in Table 2?
Comment 7) Please check all text for typos (sentences should have end, line 58; 62; 77), E. coli should be in italic; NH2 and NO2 in subscript and format references.
Author Response
Dear Reviewer,
Thank you very much for the thorough review and valuable comments. Please find bellow our response to your comments.
Reviewer 4
It is no doubt an interesting issue, and in the scope of the journal, however in my opinion, authors should make some reorganization before publication.
Comment 1) Introduction should not contain results. Data presented in Table 1 and 2 should be relocated to the section 3. Results in a subtitle 3.x. Catalytic Synthesis (for instance, or similar).
Response
We are grateful for this suggestion. We fully agree with the Reviewer. According the Reviewer suggestion Introduction was modified and Results were supported with discussion regarding synthesis of the target compounds.
Reviewer 4
Comment 2) Scheme 1, mentioned in line 80, do not appear in the text.
Response
We are grateful for this suggestion. The manuscript was revised and corrected.
Reviewer 4
Comment 3) Rewrite the sentence of lines 81 -83.
Response
We are grateful for this suggestion. The mentioned sentence was rewritten.
Reviewer 4
Comment 4) Please verify the use of the word pyridine and pyrimidine all along the text. I believe that sometimes are improperly used.
Response
We are grateful for this suggestion. The manuscript was revised and corrected.
Reviewer 4
Comment 5) Why authors used 100 % of organic solvent, as reaction medium? Under these conditions, it may be expected low yields due to enzyme inactivation.
Response
We are grateful for this suggestion. The promiscuity of enzymes is related to their unnatural activity also in organic solvents. This phenomenon is well recognized (A M Klibanov, Nature. 2001 Jan 11; 409 (6817): 241-6. doi: 10.1038 / 35051719). We conducted research on the use of various solvents, including water, and the best results were obtained when 100% of organic solvent was used.
Reviewer 4
Comment 6) Why present melting point values in Table 2?
Response
Due to the fact that the measurement of the melting point is the easiest way to confirm the structure of the obtained compound as well as to assess its purity, we decided to make this parameter readily available to the reader. Measured melting points were in agreement with the literature data
Reviewer 4
Comment 7) Please check all text for typos (sentences should have end, line 58; 62; 77), E. coli should be in italic; NH2 and NO2 in subscript and format references.
Response
We are grateful for this suggestion. We made our best efforts to improve manuscript. We improved references, literature format and its numbering

Round 2
Reviewer 1 Report
Authors have addressed the different points and now the article is fully publishable.
Author Response
We are very grateful to the Reviewer for the effort put into the evaluation of our work. Thanks to this, our work has been significantly improved.

Reviewer 4 Report
Although I consider that the manuscript was improved by the changes implemented by the authors, in my opinion, the Conclusions, of the second version, still need to be rewrite. I suggest that authors do not use topics. Please write the conclusions in plain text and use impersonal expressions (Authors should avoid the expression: In our experiments).
Author Response
Reviewer 2
Although I consider that the manuscript was improved by the changes implemented by the authors, in my opinion, the Conclusions, of the second version, still need to be rewrite. I suggest that authors do not use topics. Please write the conclusions in plain text and use impersonal expressions (Authors should avoid the expression: In our experiments).
Response
We are very grateful to the Reviewer for the effort put into the evaluation of our work. Thanks to this, our work has been significantly improved. According to the Reviewer suggestion the Conclusions were revised and modified.
